# Topology-Embedded Temporal Attention for Fine-Grained Skeleton-Based Action Recognition

Pengyuan Han [1], Zhongli Ma [2] and Jiajia Liu [2,*]

1   College of Electronic Engineering, Chengdu University of Information Technology, Chengdu 610103, China
2   College of Automation, Chengdu University of Information Technology, Chengdu 610103, China
*   Correspondence: liujj@cuit.edu.cn

**Abstract:** In recent years, graph convolutional networks (GCNs) have been extensively applied in numerous fields, demonstrating strong performances. Although existing GCN-based models have extraordinary feature representation capabilities in spatial modeling and perform exceptionally well in skeleton-based action recognition, they work poorly for fine-grained recognition. The key issue involves tiny distinctions between multiple classes. To address this issue, we propose a novel module named the topology-embedded temporal attention module (TE-TAM). Through embedding the temporal-different topology modeled with local area skeleton points in spatial and temporal dimensions, the TE-TAM achieves dynamical attention learning for the temporal dimensions of distinct data samples, to capture minor differences among intra-frames and inter-frames, making the characteristics more discriminating, and increasing the distances between various classes. To verify the validity of the proposed module, we inserted the module into the GCN-based models and tested them on FSD-30. Experimental results show that the GCN-based models with TE-TAMs outperformed the property of pred GCN-based models.

**Keywords:** topology embedding; temporal attention; GCNs; fine-grained recognition

## 1. Introduction

With the rapid development of depth sensors and skeleton extraction algorithms in recent years [1,2], skeleton data volumes have increased rapidly, so skeleton-based action recognition has received increasing attention [3–19]. Initially, skeleton data were processed into sequences [8] or pseudo-images [15], and then fed into the network for prediction, because of the advancements in recurrent neural networks and convolutional neural networks. However, in this way, the topology information in skeleton data was inevitably ignored. Graph convolutional networks demonstrate exceptional ability in processing non-Euclidean data; a plethora of new and improved graph convolutional networks have emerged to process the skeleton data, yielding promising results [3–5]. Therefore, skeleton-based action recognition is widely used in a variety of fields, including security monitoring [20], sports competitions [21], competitive training [22], robotics [23], and so on. However, some classical graph convolutional networks [3–5] still have some shortcomings when it comes to fine-grained skeleton-based action recognition.

We analyzed the network structures in some recent works, such as STGCN [3] by Sijie Yan et al., AGCN [4] by Lei Shi et al., and CTRGCN [5] by Yuxin Chen et al. We discovered that they all have a common feature: they actively improve the topology structure modeling spatially and do not place as much emphasis on the temporal dimensions. However, the difficulty in fine-grained action recognition based on skeleton data lies in the small action changes among temporal dimensions, which may only be a few frames between different actions, resulting in large intra-class variances and small inter-class variances among data samples, making the recognition extremely difficult. Therefore, we believe the key to solve the problem is that the model can capture the differences between several frames of different categories, further making the features more obvious and distinguishable, improving

the accuracy of fine-grained action recognition. Although some new methods for modeling the temporal dimensions are proposed, for example, by capturing long-term context information [8] or modeling inter-frame relationships in the temporal dimensions [12,17], they are still insufficient in fine-grained action recognition and are not for GCN-based models.

Obtaining ideas from fine-grained image classification and the attention mechanism, in this paper, we propose a novel module named topology-embedded temporal attention, which can achieve dynamic and effective temporal attention learning of skeleton data samples end-to-end (as shown in Figure 1) by embedding the topology–temporal relationship modeled with local area skeleton points in spatial and temporal dimensions, and then refine the input features with learned attention. Our proposed TE-TAM first introduces two mutually independent functions to raise the dimensions of input features, and then establishes the dependency relationship of input features in local areas (both temporal and spatial dimensions). Following that, by performing a matrix multiplication operation on the exporting dependency relation, topology relationship data in temporal dimensions are then obtained, which contain more discriminative data than only modeling in spatial dimensions, as they include not only the spatial topology relationships among bone points in a specific frame, but also the topology relationships among bone points in adjacent frames in the same sampling space. The topology relationship information is then aggregated and embedded as the excitation features of attention. After that, multi-layer perceptron is utilized to transform and activate the excitation features, to obtain the attention of the sample in the temporal dimension and capture the small movement changes among similar movements. Finally, from the perspective of signal processing, we add a residual connection [19] branch to the TE-TAM, introducing a small number of parameters to perform a simple transformation on the input and then fuse with the output (thereby restricting the output signal). To some extent, our proposed module can make up for the deficiency of the existing GCN-based models: it can capture the difference off skeleton data in the temporal dimension, to improve the performance of the existing GCN-based model network in the fine-grained skeleton-based action recognition.

Inserting TE-TAM into existing GCN-based models, we conducted a large number of experiments on FSD-30; the proposed TE-TAM achieved remarkable performances and the experimental results demonstrated the effectiveness of the TE-TAM. Simultaneously, based on the experimental results, we carefully analyzed the factors influencing the performance of each basic unit in the TE-TAM module.

The main contributions of our work are as follows:

- The proposed topology-embedded temporal attention module can construct richer topology relationship information by modeling local–area bone points (spatially and temporally), for effective attention learning, to refine input features from end-to-end, capturing minor action differences (temporally) in this way, and compensating for the shortcomings of existing GCN-based models.
- Experimental results verify the efficacy and robustness of the TE-TAM and prove the importance of simultaneous modeling of local area dependencies between adjacent bone points in spatial and temporal dimensions for attention learning. TE-TAMs inserted into existing GCN-based models outperformed the pred GCN-based methods (testing on FSD-30).

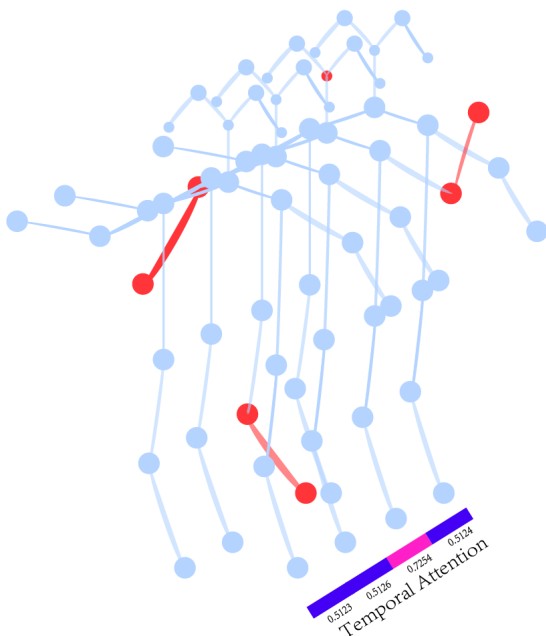

**Figure 1.** Topology-embedded temporal attention. Different-colored bars represent the responses of the topology relationships of the skeleton data on each frame in the time dimension. Thus, the final feature is more distinct for classification.

## 2. Related Work

### 2.1. Skeleton-Based Action Recognition

Researchers have conducted a lot of work on skeleton-based action recognition based on the three main lines of RNNs, CNNs, and GCNs.

Given that RNNs are able to model well the long-term context-dependent information of time series, they were first used by Yong Du et al. [8] for this task. Based on this work, some innovative work was derived. For example, to pay different levels of attention to the outputs of different frames, Sijie Song et al. [13] proposed an end-to-end spatial and temporal attention model based on RNNS, which achieved attention learning with the distinguishing joints of the bones in each input frame; In the work by Chenyang Si et al. [9], SR-TSL was proposed to capture the high-level spatial structure information within each frame and the detailed temporal dynamics of the modeled skeleton sequence. To process data with noise to enhance the effectiveness of learning sequences, the work [10] by Jun Liu et al. introduced a new gating mechanism for LSTMs. A novel attention-enhanced graph convolutional LSTM network (AGC-LSTM) was proposed in [7], which could effectively extract discriminative spatiotemporal features.

CNNs were first applied to skeleton-based action recognition in the work by Chao Li et al. [15]; then, Zhiwu Huang et al. integrated Lie groups into a deep network architecture to complete action recognition [20].

The spatiotemporal graph convolutional network was first proposed to model skeleton data by Sijie Yan et al. [3]. Then, some creative improvements were derived based on this work. For example, to improve the flexibility of constructing graph structures and make full use of skeleton information, Lei Shi et al. made improvements to GCN and proposed new network architecture, 2s-AGCN [4]. Yansong Tang et al. [12] used depth progressive reinforcement learning, extracting the most informative frames for GCN; to adaptively learn the intrinsic higher-order connections of skeleton joints, Bin Li et al. [14] proposed the STGR scheme, which could be seamlessly integrated into graph convolution networks. Introducing self-attention to action recognition [15], three different self-attention networks (SAN V1, V2, V3) were proposed to describe long-range associations or capture long-term spatiotemporal relationships, improving the ability to extract high-level semantics. To enhance the performance of graph convolution, in ContextAware graph convolution [16],

the author proposed a context-aware graph convolution network (CA-GCN) to model long-range dependencies between joint points in spatial dimensions. In [17], a spatiotemporal transformer network was introduced to understand intra-frame interactions between different body parts and inter-frame correlations.

In addition to the above three mainstream methods, there were some other works, such as semantic-guided [24], which combined GCN and CNN to build a simple and effective semantic-guided neural network and PoseC3D [25]; it relies on a 3D heat map stack as the basic representation of the human skeleton.

Summarizing the above research process, it is not difficult to see that the improvements of related works around RNNs, GCNs, and other works, involved different methods to continuously strengthen the modeling abilities of the networks in the spatial and temporal dimensions, to better express the data samples. In our work, we also abided by such a principle and had a similar purpose to [12,13] (in that the model pays more attention to the temporal dimension). However, the difference is that we used the method of topology-embedding to gain attention; our module can be plugged into GCN-based models and improve their performances.

*2.2. Fine-Grained Classification*

Fine-grained classification is a popular task, particularly in image classification. Relational research on the task could be divided into the following directions: (1) using deep convolution networks with strong representation abilities, Tsung-Yu et al. firstly proposed bilinear models [26]. With two deep convolution networks as feature extractors, the model performed strongly because of the representation ability of the deep convolution networks. (2) Localization and recognition methods. In the work by Ze Yang et al. [27], they proposed an NTS-Net, consisting of a navigator agent, a teacher agent, and a scrutinizer agent. The navigator agent finds the most informative regions, and the teacher agent evaluates the regions and then provides feedback, finishing the localization. Final regions are then put into the scrutinizer to extract fine-grained features for recognition; Ning Zhang et al. [28] shared a similar idea with Ze Yang et al. [27]. In their work [28], they propose part-based R-CNNs. The network firstly proposes and scores the regions by a detector, and then extracts features on the localized regions for fine-grained recognition, as do Di Lin et al. in their work [29]. (3) Approaches based on visual attention mechanisms. In the work by Jianlong Fu et al. [30], RA-CNN is proposed. In their paper [31], Ming Sun et al. proposed a novel attention-based convolutional neural network. With a one-squeeze multi-excitation (OSME) module, the network first learned about the multiple attention region features of each input image, and then the features were utilized for recognition. A diversified visual attention network (DVAN) was proposed by Bo Zhao et al. [32]. The network contains four parts: attention canvas generation, CNN feature learning, diversified visual attention, and classification. The first three parts are for localization and the last part is for recognition. In conclusion, attention mechanisms are for localization to some extent.

To summarize the methods discussed above. It is easy to see that they all share the same goal: finding and magnifying fine-grained features to make data samples more discriminative; this concept has also been applied in our work. However, due to the different data forms, fine-grained features of skeleton-based data samples with higher dimensions than image data not only show intra-frame action differences in spatial dimensions but also show inter-frame action differences in the temporal dimensions. Thus, to solve this, our method can be used to model the topology relationship between bone points in the local area as embedded attention, which can simultaneously pay attention to action differences between and within frames.

## 3. Methods

In this section, we first introduce the topology-embedding method, and then show how the embedded excitation features are transformed into high-level representations in

the temporal space as temporal attention. Finally, we demonstrate the architecture of a basic TE-TAM block as well as the GCN-based models with TE-TAM.

### 3.1. Topology-Embedded Temporal Attention

As shown in Figure 2, we provide a straightforward topology-embedded temporal attention framework. To obtain the topology relationships of skeleton data for each frame or multiple neighboring frames and bone points, we first modeled the input feature in both spatial and temporal dimensions. The derived topology relationship was then embedded as the attention excitation features. Following that, the excitation features were converted and transferred to the time dimension space. Finally, the learned attention was applied to modify the input feature, which was then fused with the transformed input feature to produce the final output. Specifically, the TE-TAM is composed of three parts: (1) functions $\Phi(\cdot)$ and $\Omega(\cdot)$ are used to finish the topology modelings of bone points while function $\theta(\cdot)$ is used to embed the topology relationship. (2) Attention transformation is conducted by functions $\delta(\cdot)$ and $\gamma(\cdot)$, which transform the excitation features to the temporal space. (3) The residual connection feature is generated by function $\Omega(\cdot)$ with an input feature. Therefore, the TE-TAM is formulated as

$$Out = \gamma(\sigma(\theta(\Phi(X) \bigotimes \Omega(X)))) \bigodot X + \Psi(X) \tag{1}$$

Given an input of $X \in \mathbb{R}^{C \times T \times M}$, the output is $Out \in \mathbb{R}^{C \times T \times M}$, which is refined by learned attention and constrained by a residual connection.

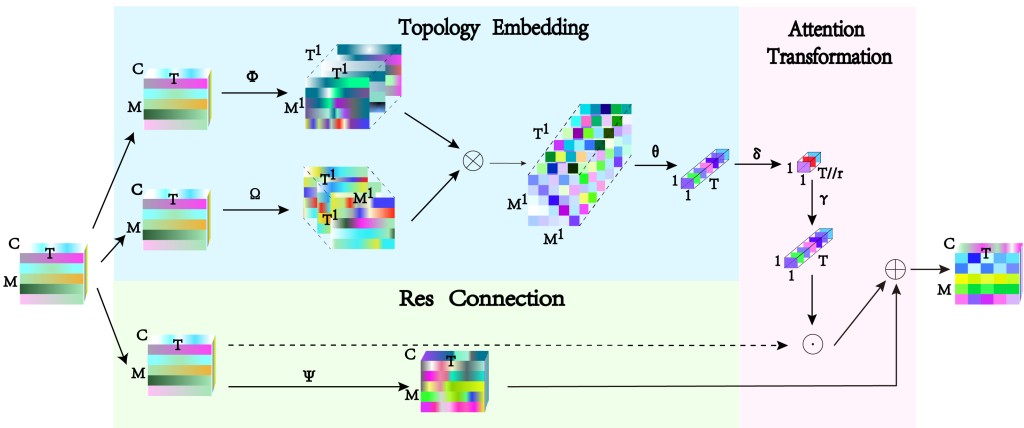

**Figure 2.** The structure of the proposed topology-embedded temporal attention module. Topology embedding is aimed at modeling the effective topology relationship, embedding as excitation features. Attention transformation completes the transformation of the excitation to the attention score. The goal of the residual connection is to restrict the output by the transformed input.

The topology-embedded temporal attention framework is introduced in detail as follows:

***Topology Embedding***. The embedding process is shown in the blue block in Figure 2. We first employed two independent functions to expand the dimensions of the input features along channel C, while modeling the dependent relationships between the bone points in both spatial and temporal dimensions. The modeling method (functions $\Phi(\cdot)$ and $\Omega(\cdot)$) are formulated as:

$$f_{out}(M_{it_s}) = \sum_{M_{jt_l} \in N_{it_s}}^{K_M} \frac{1}{Z_{jt_s jt_l}} f_{in}(M_{jt_l}) \cdot \omega \cdot (l_{it}(M_{jt_l})) \tag{2}$$

where $M_{it_s}$, $M_{jt_l}$ denote the $i_{th}$, $j_{th}$ bone point data in the temporal dimension of frame $t_s$ ($t_s$ is a set, $t_s = \{s = 1, \ldots, T\}$, if $s = 1$, $t_1 = \{frame_1\}$, if $s = 2$, $t_2 = \{frame_1, frame_2\}$; thus, the input bone point subset extends to the temporal dimension), while $N_{it_s}$ is a bone points

subset, consisting of the neighbors of $M_{it_s}$. Moreover, $K_M$, which is similar to the kernel size of the convolution operator, decides the size of the subset. The normalizing term $Z_{jt_s jt_l}$ = $|\{M_{kt_l} | l_{it}(M_{kt_l}) = l_{it}(M_{jt_l})\}|$ involves balancing the contributions of different subsets to the output. Moreover, $\omega$ is a weighting function to assign weights for bone points that are spatially and temporally adjacent. $l_{it}$ is a mapping function, which aims to expand the dimensions of each bone point of the input features.

The given formula is a generalized version. We can change the settings of the function to express different functionalities. For example, if we set the $t_s = 1$ and $K_M = 1$, as shown in Figure 3A, the function will only model a single bone point in each frame. If we set $t_s = 2$ and $K_M = 1$, demonstrated in Figure 3B, the dependencies of the bone points between the adjacent frames will be modeled. As we set $t_s = 1$ and $K_M = 2$, indicated in Figure 3C, the correlation of the neighboring bone joints will be modeled. To model the connections of the bone points, in the temporal and spatial dimensions, we set $t_s = 2$ and $K_M = 2$. Although the function was limited to modeling neighboring elements, the capacity to represent both the temporal and spatial dimensions is adequate for topological relational embedding. Specifically, in our experiment, the formula was achieved by the convolution operator. Moreover, the influence of the formula settings is shown in Section 4.

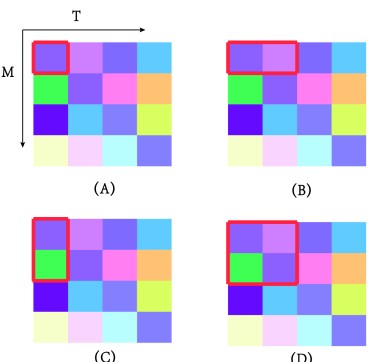

**Figure 3.** Schematic diagram with different $K_M$; (**A**) with $K_M = 1$, which only raises the dimensions of the bone points on each frame; (**B**–**D**) with different $K_M$ settings, which model the relationships of the bone points in local areas of spatial and temporal dimensions.

Based on the topology relationship modeled by the above formulation: the output $(F_{topology})$ of functions $\Phi(\cdot)$ and $\Omega(\cdot)$ are all in the space of $\mathbb{R}^{T^1 \times T^1 \times M^1}$. We first transposed one of them to $\mathbb{R}^{T^1 \times M^1 \times T^1}$, and then multiplied them. Thus, the output of the matrix multiplication is in the space of $\mathbb{R}^{T^1 \times M^1 \times M^1}$. Moreover, it was used as the representation of the topology relationship among the temporal dimensions. Finally, the topology relationship was embedded as excitation features of attention with function $\theta$. The function is formulated as:

$$F_{embedded} = \begin{cases} AvgPool(F_{topology}) \\ MaxPool(F_{topology}) \end{cases} \tag{3}$$

The primary goal of feature embedding is to gather and capture effective data in topological relations, as well as to locate the features that best characterize the topological relations. Only in that way, could they be used as effective excitation features of attention. Therefore, we employed average pooling to aggregate topological information or used max pooling to capture the distinction on each frame, and at the same time reduce the feature dimensions without introducing any parameters to the network structure. The experimental findings of the performances of the two distinct pooling approaches on the verification set are shown in Section 4.

***Attention Transform***. The embedded feature $F_{embedded}$ in $\mathbb{R}^{T^1 \times 1 \times 1}$ is first reshaped to $\mathbb{R}^{T^1}$. Moreover, it is transformed to $\mathbb{R}^T$ with functions $\delta(\cdot)$ and $\gamma(\cdot)$, formulated by:

$$A_{attention} = \sigma(MLP(F_{embedded})) \tag{4}$$

Functions $\delta(\cdot)$ and $\gamma(\cdot)$ are critical to the overall TE-TAM since they are intended to turn excitation features into attention. Therefore, we employed the multilayer perceptron (MLP), which contained powerful fitting abilities. Moreover, there were several choices for the activation functions $\sigma(\cdot)$ and $\sigma_1(\cdot)$ in MLP, such as sigmoid, ReLU, and tanh. We chose ReLU and sigmoid as the two activation functions according to our experiments in Section 4.

***Residual Connection***. As shown in the cyan block in Figure 2, the third branch of the input feature is a very simple operation and works efficiently. Moreover, it is formulated as:

$$f_{out} = \Psi(f_{in}) \tag{5}$$

To improve the performance of the framework and constraint the output feature, we added the residual connection to our framework. In addition, as the output and input does not change the distribution of the data, we used the residual connection with function $\Psi$, which provided a parameter that was trained end-to-end, to adjust the output dynamically.

### 3.2. Topology-Embedded Temporal Attention Block

The implementation details of our proposed topology-embedded temporal attention are shown in Figure 4. We used $Conv3 \times 3$ to simultaneously model the dependencies among local bone points of input features in the temporal and spatial dimensions. Therefore, the topology relationships contained more information. Then, $Avgpool$ was applied to aggregate the topology relationship in the spatial dimensions and retain the temporal dimensions. Moreover, the aggregations were as the excitation features; they were transformed to 'attention' using the *Linear* layer and *activation* functions. Finally, the modified input features with attention were fused with the $Conv1 \times 1$-transformed input features to realize the block.

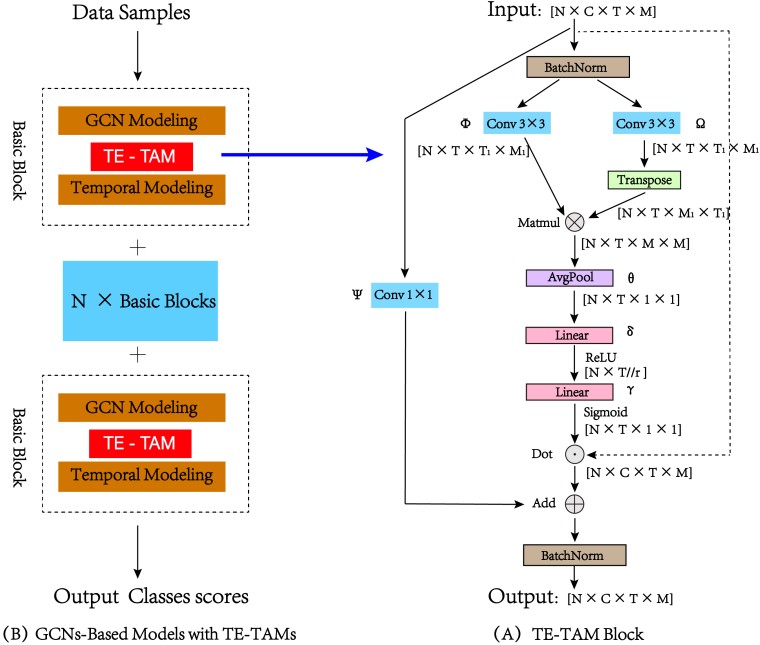

**Figure 4.** The implementation of our proposed TE-TAM. $Conv3 \times 3$ completes the topology relationship modeling; $Avgpool$ finishes the topology embedding; the *Linear* layer and activation functions transform and activate the excitation features while $Conv1 \times 1$ transforms the input features.

### 3.3. Existing GCNs with TE-TAM

In combination with our proposed TE-TAM, we made some improvements into existing GCN-based models. Through the detailed structural analysis of classic graph convolution networks in recent years (STGCN [3], AGCN [4], CTRGCN [5]), we found that the architecture of the basic blocks were composed of two basic units: spatial modeling with graph convolution and temporal modeling. The function of the TE-TAM was to refine the input features through attention to the temporal dimension. Therefore, the TE-TAM was inserted between spatial modeling and temporal modeling as a basic unit, forming the basic blocks. These basic blocks were then stacked to form a new network architecture, as shown in Figure 4.

## 4. Experiments

### 4.1. Dataset

The figure skating dataset with 30 classes (FSD-30) [33] was used in our experiments. The dataset was created using video footage in figure skating competitions from 2017 to 2022. To maintain the relative consistency of the dataset, the frame rate of the video in the source video material was standardized to 30 frames per second, and the image size was standardized to $1080 \times 720$. Following that, the Open Pose 2D Pose estimation algorithm [1,2] was used to extract the bone points of the video, frame-by-frame. Moreover, the dataset was stored in *npy* format. The dataset included 30 classes and 4184 samples, with each sample consisting of bone point data from successive frames of action. All classes of samples were randomly and evenly divided into a training set of 2922 samples and two test sets of 628 and 624 samples, respectively. Moreover, we named the three sets *TrainSet$_1$*, *TestSet$_1$*, and *TestSet$_2$*.

### 4.2. Training Details

All of our experiments were performed on the Tesla V100 computing platform using the PaddlePaddle [34] deep learning framework. The momentum optimization method was used; momentum was set to 0.9. The batch size was set to 32. Moreover, the mixup data augmentation strategy was employed while training and $\alpha$ was set to 0.2. The cross-entropy function was applied as the loss function to the backpropagation gradient and the updated parameters.

In addition, to test the TE-TAM on GCN-based models (STGCN [3], AGCN [4], CTRGCN [5]), different learning rate decay strategies were used: (1) CTRGCN [5] inserted by the TE-TAM used the customwarmup cosinedecay method to dynamically adjust the learning rate. Moreover, the parameters *max epoch*, *warmup epochs*, *warmup start learning rate*, and *cosine base learning rate* were set to 65, 10, 0.005, and 0.05. (2) AGCN [4] inserted by the TE-TAM also used the customwarmup cosinedecay method. Moreover, the above parameters were set to 100, 50, 0.005, and 0.05. (3) STGCN [3] inserted by the TE-TAM used the cosine annealing decay method. The *learning rate* and $T_{max}$ were each set to 0.05 and 90.

Finally, it should be noted that the dataset was processed in different ways for the ablation study and comparison among GCN-based models. For experiments in the ablation study, each datum sample with the max number of frames in the dataset was set to 350, to analyze the factors affecting the performance of each unit in the module as soon as possible.

### 4.3. Ablation Study

In this section, we explore the influences of different settings in the proposed TE-TAM module on the performance through experiments. Note: all experiments in this section were performed on *TrainSet$_1$* with CTRGCN [5] + TE-TAM. We randomly and evenly divided 15% of the sample data of all categories in *TrainSet$_1$* into the test set (named *TestSet$_3$*), and the rest into the training set (*TrainSet$_2$*); all of the models were trained on *TrainSet$_2$* and tested on *TestSet$_3$*; they share the same training settings as mentioned in Section 4.2. In addition, the baseline is the result of CTRGCN.

As shown in Table 1, if we number the ten blocks in the model from one to nine: 0123456789, then, the insertion positions of TE-TAM in the ABCDEFH experiments are: 01, 09, 0123, 0189, 012345, 012789, and 479.

We conducted relevant experimental explorations on the effectiveness of the TE-TAM module. By comparing the experimental results from inserting different numbers of TE-TAM modules and inserting the TE-TAM in different positions, we proved the effectiveness of the TE-TAM and drew the following conclusions: (1) only when the TE-TAM is placed in an appropriate position can it play an effective role; (2) when the temporal dimensions of the input features decrease, the effect of the TE-TAM will weaken, accordingly. Specifically, by comparing A and B, we change the position of the A module, and the accuracy of experiment B improves compared with that of experiment A. Meanwhile, comparing their results with that of experiment H, we find that the position of the TE-TAM has a relatively great influence on the performance. Through the comparisons of E and F, and E and G, combined with the experimental results of A, B, C, and D, it can be concluded that when the temporal dimensions of the feature are low, the effects of TE-TAM will be reduced.

**Table 1.** Comparison of Top-1 Acc with blocks of different stacked numbers and inserted positions.

| Methods | Top-1 Acc (%) |
|:---:|:---:|
| **Baseline** | **59.8** |
| +2 *TE-TAM blocks A* | $56.6^{\downarrow 3.2}$ |
| +2 *TE-TAM blocks*[1] *B* | $58.9^{\downarrow 0.9}$ |
| +4 *TE-TAM blocks C* | $59.6^{\downarrow 0.2}$ |
| +4 *TE-TAM blocks*[1] *D* | $58.9^{\downarrow 0.9}$ |
| +6 *TE-TAM blocks E* | $60.7^{\uparrow 0.9}$ |
| +6 *TE-TAM blocks*[1] *F* | $\mathbf{61.4}^{\uparrow 1.6}$ |
| +10 *TE-TAM blocks G* | $\mathbf{61.4}^{\uparrow 1.6}$ |
| +3 *TE-TAM blocks*[2] *H* | $\mathbf{60.2}^{\uparrow 0.4}$ |

*blocks*: TE-TAMs are stacked from top to bottom. *blocks*[1]: TE-TAMs are inserted from both sides of the model. *blocks*[2]: TE-TAMs are inserted in special places. Note: Other parameters are set: $K_M = (1,1)$, $\theta = \theta_1$, $r = 2$, $\sigma = sigmoid$ and $\sigma_1 = relu$.

Through the horizontal and vertical comparative analyses of the experimental results as shown in Table 2, it can be found that: (1) only modeling the topological relations of the inner skeleton data in the single frame can have limited attention, which also conforms to the fact that the action features of adjacent and similar frames are similar; (2) it is effective to model the topology relationships of local bone points adjacent to the temporal and spatial dimensions, simultaneously. In addition, the effect is most obvious when $K_M = (3,3)$.

**Table 2.** Comparison of Top-1 Acc with different settings of the $K_M$ size.

| $K_M$ Size | Top-1 Acc (%) | $K_M$ Size | Top-1 Acc (%) |
|:---:|:---:|:---:|:---:|
| baseline | 59.8 | (1, 3) | $\mathbf{60.3}^{\uparrow 0.5}$ |
| (1, 1) | $\mathbf{61.4}^{\uparrow 1.6}$ | (3, 1) | $\mathbf{61.6}^{\uparrow 1.8}$ |
| (1, 2) | $\mathbf{60.7}^{\uparrow 0.9}$ | (2, 3) | $\mathbf{60.7}^{\uparrow 0.9}$ |
| (2, 1) | $58.7^{\downarrow 1.1}$ | (3, 2) | $\mathbf{60.0}^{\uparrow 0.2}$ |
| (2, 2) | $\mathbf{60.5}^{\uparrow 0.7}$ | (3, 3) | $\mathbf{61.8}^{\uparrow 2.0}$ |

Note: Other parameters are set: $\theta = \theta_1$, $r = 2$, $\sigma = sigmoid$ and $\sigma_1 = relu$; as well as with ten TE-TAM blocks.

We explored the influences of the different settings of the embedding functions, activation functions, and parameter *r* on the results. As shown in Table 3, I and J show that different embedding functions have a great impact on the results, which also indirectly illustrates the importance of topology embedding. Experiments K, L, M, N, O, and P illustrate that the different configurations of the MLP can affect the conversion of the excitation

features obtained by topology embedding. At the same time, the importance of the MLP is also indirectly proved.

**Table 3.** Comparison of Top-1 Acc with different parameter settings of $\theta$, r, $\sigma$, $\sigma_1$.

| Methods | $\theta$ | r | $\sigma$ | $\sigma_1$ | Top-1 Acc (%) |
|---|---|---|---|---|---|
| **Baseline** | - | - | - | - | **59.8** |
| $I$ | $\theta_1$ | 2 | sigmoid | relu | **$61.8^{\uparrow 2.0}$** |
| $J$ | $\theta_2$ | 2 | sigmoid | relu | $57.8^{\downarrow 2.0}$ |
| $K$ | $\theta_1$ | 4 | sigmoid | relu | $54.6^{\downarrow 5.2}$ |
| $L$ | $\theta_1$ | 8 | sigmoid | relu | $58.5^{\downarrow 1.3}$ |
| $M$ | $\theta_1$ | 2 | sigmoid | tanh | $59.4^{\downarrow 0.4}$ |
| $N$ | $\theta_1$ | 2 | sigmoid | sigmoid | $56.4^{\downarrow 3.4}$ |
| $O$ | $\theta_1$ | 2 | relu | relu | $59.8^{-}$ |
| $P$ | $\theta_1$ | 2 | tanh | relu | $57.8^{\downarrow 2.0}$ |

Note: Parameter is set: $K_M = (3, 3)$ and with ten TE-TAM blocks.

### 4.4. Visualization of Learned Attention

The attention learned by the TE-TAM is visualized to directly demonstrate the effectiveness of the TE-TAM. As different layers of the network are not consistent in the temporal dimensions, for display convenience, we only showed the attention of the first thirty frames of all layers in the network. Figure 5 shows the learned attention of the TE-TAM at each layer. The attention learned by layers 5, 6, 7, 8, and 9 was relatively more discriminative in the first thirty frames. The attention score ranged from 0 to 1. The closer the value was to zero, the less weight was assigned to the frame, and the frame's features were weakened. The closer the value was to one, the more weight was assigned to the frame, and the features within the frames were magnified. Small differences in the temporal dimensions can thus be captured, making features more discriminative.

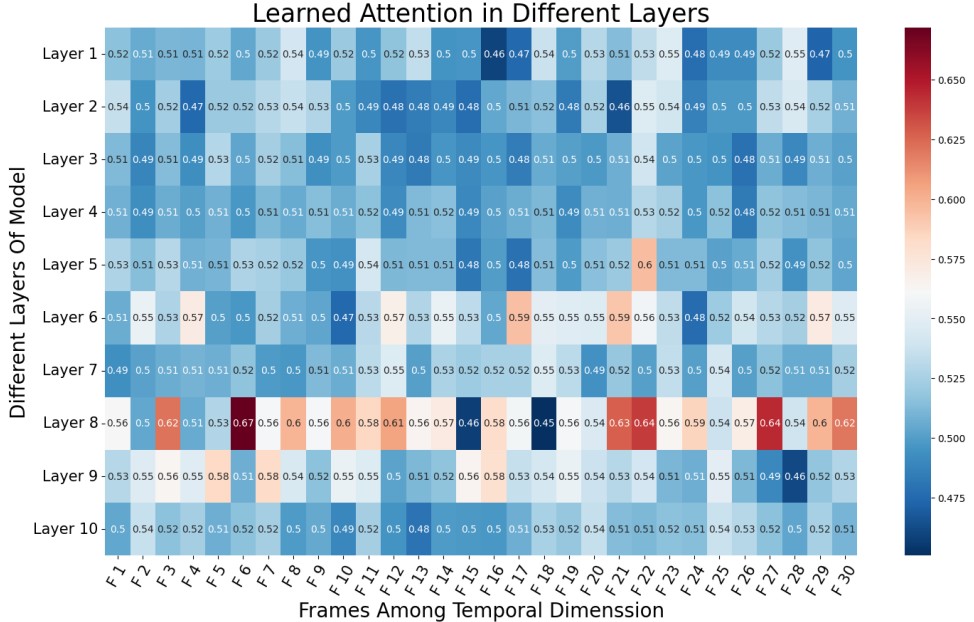

**Figure 5.** Visualization of learned attention by CTRGCN paired with TE-TAMs in different layers. We only show thirty frames of the first data sample in *TestSet*$_3$.

### 4.5. Comparison among GCN-Based Models

We trained AGCN, STGCN, and CTRGCN with and without TE-TAMs using the parameters mentioned in Section 4.3 on *TrainSet*$_1$. There were no tricks used in the training process to demonstrate the validity of TE-TAM. Moreover, the Loss, Top-1 accuracy, and Top-5 accuracy of each model during training is depicted in Figure 6. As shown in Table 4, we tested each of the above models on *TestSet*$_1$. The effectiveness of TE-TAM was demonstrated by comparing the Top-1 Acc(%)[1]. Furthermore, TE-TAM could be inserted into various GCN-based models to improve their performances. Thus, the universality and robustness of TE-TAM are demonstrated to some extent.

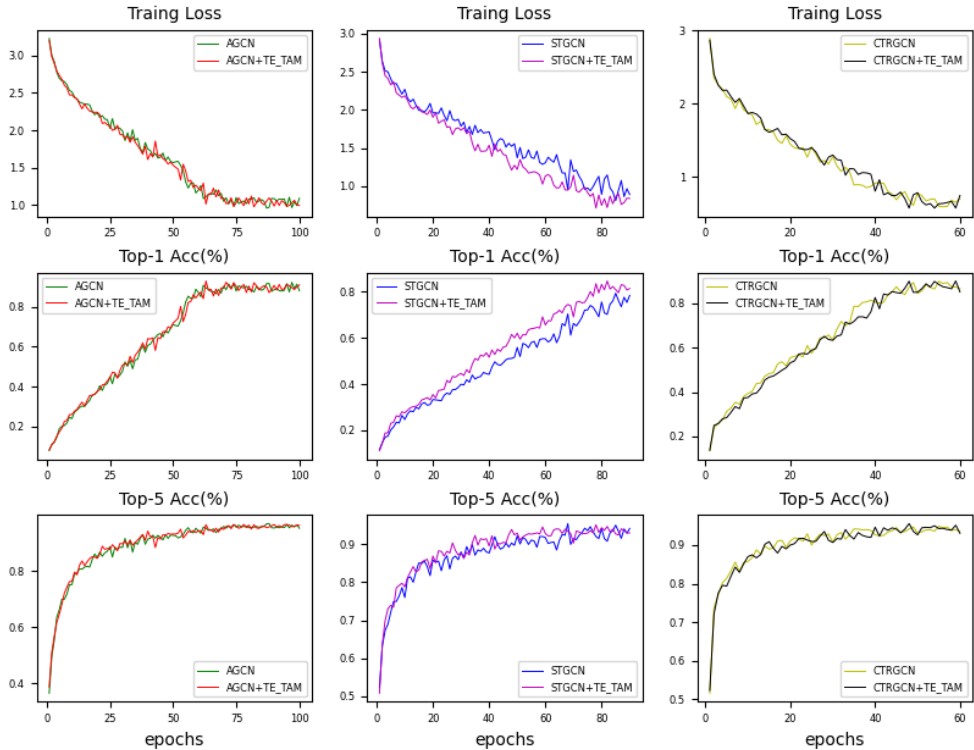

**Figure 6.** Training process. Training loss, Top-1 Accuracy, and Top-5 Accuracy of AGCN, STGCN, and CTRGCN with and without TE-TAM.

**Table 4.** Comparison of accuracy for three GCN-based models with and without TE-TAM modules.

| Methods | Top-1 Acc (%)[1] |
|:---:|:---:|
| STGCN [3] | 60.8 |
| STGCN+TE-TAMs | **61.2**$^{\uparrow 0.4}$ |
| AGCN [4] | 58.9 |
| AGCN+TE-TAMs | **60.2**$^{\uparrow 1.3}$ |
| CTRGCN [5] | 61.3 |
| CTRGCN+TE-TAMs | **63.1**$^{\uparrow 1.8}$ |

Top-1 Acc(%)[1]: Top-1 Acc on *TestSet*$_1$.

## 5. Conclusions

In this work, we propose a novel topology-embedded temporal attention module (TE-TAM) to improve the performance of GCNs for fine-grained skeleton-based action recognition. GCN-based models with TE-TAMs achieve dynamic attention learning by embedding the informative and flexible topology relationship constructed by modeling local areas in spatial and temporal dimensions. In addition, experiments on FSD-30 proved the importance of simultaneous modeling of local area dependencies between adjacent

bone points in spatial and temporal dimensions for attention learning and the effectiveness/robustness of the TE-TAM. Finally, we should acknowledge the weaknesses of the proposed TE-TAM: (1) the topology modeling method in our experiments introduced much more parameters but showed a relatively weak performance; (2) the embedding function chosen in this work was relatively simple and rough. Thus, the TE-TAM may have not yet been fully functioning. We will address the problem in our future work.

**Author Contributions:** Formal analysis, Z.M.; Methodology, P.H.; Resources, Z.M.; Software, P.H.; Writing—original draft, P.H.; Writing—review & editing, Z.M. and J.L. All authors have read and agreed to the published version of the manuscript.

**Funding:** This research received no external funding.

**Acknowledgments:** The authors are grateful to the College of Automation, Chengdu University of Information Technology. This paper is supported by the International Cooperation Project of Science and Technology Bureau of Chengdu (no. 2019-GH02-00051-HZ), the Sichuan Intelligent Unmanned Systems–Intelligent Perception, Engineering Laboratory Open Fund, and the research fund of the Chengdu University of Information Engineering, under grant nos. WRXT2020-001, WRXT2020-002, WRXT2021-002, and KYTZ202142. This paper is also supported by the Sichuan Science and Technology program, China, grant no. 2022YFS0565.

**Conflicts of Interest:** The authors declare no conflict of interest.

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
