# Peer review of "Topology-Embedded Temporal Attention for Fine-Grained Skeleton-Based Action Recognition"

_applsci, doi:10.3390/app12168023_

Round 1

Reviewer 1 Report

The paper presents a topology embedded temporal attention module, named TE-TAM, which is expected to construct richer topology relationship information. The work is interesting, and the manuscript is clear, and shows the performance and disadvantages of this module. 

There is a number of issues that the authors should consider when revising their paper:

- Line 30 may lead to confusion. This sentence suggests that the works [3], [4] and [5] are written by the same authors than the present paper, but it does not seem so. Please clarify.

- Section 2.1 presents quite exhaustively some related works about skeleton-based action recognition. However, section 2.2 is too short. The authors should present more in depth the state of the art on fine-grained classification.

- Also, at the end of section 2, the authors should more clearly state the need of their work, and the main differences with respect to the recent, related works in the state of the art.

- Please clearly state which is the meaning of all the symbols and functions in equation (2).

- It would be nice to study the influence of omega in the results.

- The paper should be carefully proofread, as one can find a number of issues with the use of the English language.

Author Response

Dear Reviewer,   

Thank you very much for your time involved in reviewing the manuscript and your very encouraging comments on the merits.

We have provided a point-by-point response to your comments. Please see the attachment.

We would like to take this opportunity to thank you for all your time involved and this great opportunity for us to improve the manuscript. We hope you will find this revised version satisfactory.

Sincerely,

Pengyuan Han

Reviewer 2 Report

The author presented a timely research work entitled "Topology Embedded Temporal Attention for Fine-Grained Skeleton-Based Action Recognition". A huge number of researches have already been done in this area but why has the author selected this topic. The author should explain the motivation as well as contributions clearly in the abstract, introduction, methodology, result analysis, and conclusion.

The paper is technically sound in the methodology part but suffers from a lack of novelty. The methodology part is very poor in writing as well as representation and organization.  The author should compare his method with more sophisticated machine learning algorithms for feature extraction and classification like decision tree, fuzzy entropy-based analysis, transfer learning, search learning, and so on instead of using only traditional CNN.

Author Response

(The authors gave the same response as above.)

Round 2

Reviewer 2 Report

Now, it looks better from previous version after providing the proper explanations ofmy comments.